# Let's Revise Step-by-Step: A Unified Local Search Framework for Code Generation with LLMs

Zhiyi Lyu[1]        Jianguo Huang[1]        Yanchen Deng[1]*

Steven Hoi[2]                Bo An[1]

[1] Nanyang Technological University    [2] Alibaba Group

## Abstract

Large Language Models (LLMs) with inference-time scaling techniques show promise for code generation, yet face notable efficiency and scalability challenges. Construction-based tree-search methods suffer from rapid growth in tree size, high token consumption, and lack of anytime property. In contrast, improvement-based methods offer better performance but often struggle with uninformative reward signals and inefficient search strategies. In this work, we propose **ReLoc**, a unified local search framework which effectively performs step-by-step code revision. Specifically, ReLoc explores a series of local revisions through four key algorithmic components: initial code drafting, neighborhood code generation, candidate evaluation, and incumbent code updating, each of which can be instantiated with specific decision rules to realize different local search algorithms such as Hill Climbing (HC) or Genetic Algorithm (GA). Furthermore, we develop a specialized revision reward model that evaluates code quality based on revision distance to produce fine-grained preferences that guide the local search toward more promising candidates. Finally, our extensive experimental results demonstrate that our approach achieves superior performance across diverse code generation tasks, significantly outperforming both construction-based tree search as well as the state-of-the-art improvement-based code generation methods. The code is available at https://github.com/alphatogo/ReLoc.

## 1 Introduction

Large Language Models (LLMs) like GPT-4 (Achiam et al., 2023) and Claude (Wang et al., 2024a) have demonstrated remarkable capabilities in code-related tasks, including code generation (Chen et al., 2021; Austin et al., 2021), repair (Xia and Zhang, 2022; Jiang et al., 2023; Jin et al., 2023), and optimization (Shypula et al., 2023; Cummins et al., 2023). However, when facing challenging tasks, their auto-regressive token generation process prohibits the use of additional computational resources to achieve better performance (Yao et al., 2023; Snell et al., 2024). To fully unleash the power of LLMs, recent studies have focused on inference-time scaling techniques like construction-based tree-search algorithms (Feng et al., 2023; Wang et al., 2024b), which incrementally build a high-quality full response via exploring a tree of intermediate reasoning steps guided by a value model (Wang et al., 2023) or a Process-based Reward Model (PRM) (Lightman et al., 2023).

Despite their potential, construction-based tree-search methods suffer from a rapid increase in tree size and excessive token consumption as the number of reasoning steps grows, which inevitably leads to insufficient exploration given a practical budget. Besides, these methods do not hold the *anytime*

---

*Correspondence to: ycdeng@ntu.edu.sg

39th Conference on Neural Information Processing Systems (NeurIPS 2025).

property (Zilberstein, 1996), since they cannot return a response until they find a reasoning path from the root to a leaf in the search tree. In contrast, recent approaches (Li et al., 2024a; Light et al., 2024) that utilize multi-turn improvements (Zheng et al., 2024) on complete responses for code generation have shown promise. These multi-turn approaches explore a series of local revisions on the code by feeding the execution feedback from public test cases (Xia et al., 2024) back to the LLM, which mirrors how humans iteratively refine code drafts and enjoys the anytime property.

However, the existing improvement-based approaches still face significant challenges in efficiently finding high-quality codes (Olausson et al., 2023). First, many of those methods rely on *ad-hoc* reward functions (e.g., the pass rate on the public test cases or simply an LLM's self-evaluated score) to measure the code quality, which may fail to provide informative direction to guide the search process (cf. Figure 1). In more detail, since the number of public test cases is usually small (e.g., 2-3 test cases per task), the pass rate often collapse to binary signals (i.e., 0% or 100%), offering little guidance in selecting promising code revisions. LLM-based self-evaluation, on the other hand, is prone to hallucinations (Zhang et al., 2024a) and can mislead the search by inaccurately assessing a code revision's potential. Second, the inefficient code revision generation and search algorithms exacerbate the token consumption. For example, the agentic methods (Li et al., 2024a; Wang et al., 2024a) exploit complex workflows which would consume a large number of tokens even over a few improvement iterations. Besides, search algorithms like Monte Carlo Tree Search (MCTS) (Li et al., 2024b) often require a significant number of improvement iterations

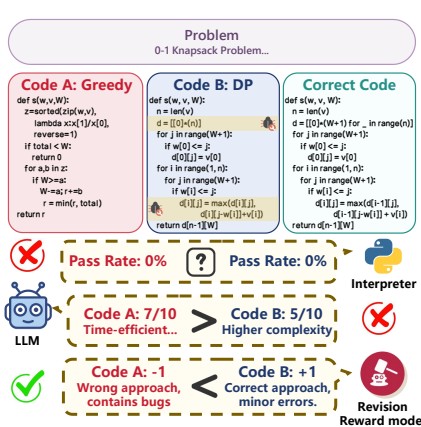

Figure 1: In the 0-1 Knapsack problem, the LLM favors an efficient but incorrect greedy solution (Code A) over a conceptually correct yet buggy DP version (Code B). As the tied pass rate offers no guidance, a revision reward model instead prioritizes candidates that are easier to revise into correct solutions.

to balance exploration and exploitation, leading to excessive computational overhead, e.g., running at least 50 iterations for comparable performance which consumes over 20,000 tokens on a single task. An extended discussion of related work is available in the Appendix.

In light of this, we introduce a lightweight, unified local search framework for improvement-based code generation with LLMs. The core idea behind our approach is to explore the neighborhood of the incumbent code with simple-yet-effective decision rules. To this end, we first frame the iterative improvement process within a local search framework with four key algorithmic components: initial code drafting, neighborhood code generation, candidate evaluation and incumbent code updating. Each component can be instantiated with a specific decision rule, allowing the development of different local search algorithms for code generation with LLMs. Furthermore, to facilitate candidate evaluation in each iteration, we develop a specialized revision reward model tailored for local search which is trained to prefer the code with a smaller *revision distance*, i.e., the minimum number of revision steps required to transform it into a corrected version. Intuitively, instead of solely maximizing the pass rate on the public test cases which sometimes can be uninformative, our reward model works directly on the textual space and guides the local search to explore the codes *close to* the correct ones. Specifically, our main contributions are summarized as follows.

1) We propose ReLoc, a lightweight and unified local search framework for code generation. ReLoc can be effectively instantiated into different local search algorithms such as Hill Climbing (HC) (Russell and Norvig, 2016) and Genetic Algorithm (GA) (Mitchell, 1998) by implementing each algorithmic component with a specific decision rule.

2) We develop a revision reward model trained with pairwise supervisions derived from revision distance comparisons. By constructing win/loss pairs based on which candidate is closer to the corrected code, we train the reward model using the Bradley–Terry framework (Bradley and Terry, 1952) to produce fine-grained preferences that guide the local search toward promising candidates, even when explicit correctness signals are uninformative.

3) We conduct extensive experimental evaluations on popular code generation benchmarks including LiveCodeBench (Jain et al., 2024) and TACO (Li et al., 2023). The results demon-

strate that our local search approaches consistently outperform both construction-based tree-search methods and existing improvement-based approaches, achieving a **33.8%→38.4%** improvement in Pass@1 on LiveCodeBench and an **11.5%→15.3%** gain on TACO over the strongest baseline, while reducing token consumption by **37%** (cf. Figure 4).

## 2 Preliminaries

**Code generation tasks.** A code generation task can be defined as a triple $\mathcal{T}_i = \langle x_i, u_i, v_i \rangle$ where where $x_i$ is the problem statement given in natural language, $u_i$ is a set of public test cases, and $v_i$ is a set of private test cases. A code sample $a$ is deemed correct if it passes all test cases in both $u_i$ and $v_i$. Given a set of code generation tasks $\mathcal{T} = \{\mathcal{T}_1, \ldots, \mathcal{T}_N\}$, the performance of a code generation policy $\pi$ is measured by Pass@$k = \mathbb{E}_{\mathcal{T}_i \sim \mathcal{T}} \left[ 1 - \frac{\binom{n-c_i}{k}}{\binom{n}{k}} \right]$, where $n$ is the total number of codes sampled for each task with policy $\pi$ and $c_i$ is the number of correct code samples for task $\mathcal{T}_i$ (Chen et al., 2021).

**Improvement-based code generation.** The improvement-based code generation process can be formalized as an episodic Partially Observable Markov Decision Process (POMDP), defined by the tuple $\langle \mathcal{S}, \mathcal{A}, \mathcal{O}, p, r, T \rangle$. The state $s_t \in \mathcal{S}$ includes the code generation task $\mathcal{T}_i$, the current code sample $a_t$, the execution feedback from public test cases $f(a_t; u)$ and the one from private test cases $f(a_t; v)$ for each time step $t$. Particularly, the initial state $s_0 = \mathcal{T}_i$. The action $a_t \in \mathcal{A}$ is a token sequence which constitutes a code sample. The state transition function $p : \mathcal{S} \times \mathcal{A} \to \mathcal{S}$ deterministically updates the state by evaluating the code sample $a_{t+1}$, yielding $s_{t+1}$. The observation $o_t \in \mathcal{O}$ is a subset of $s_t$, consisting of $\langle x_i, u_i \rangle$, $a_t$ and $f(a_t; u)$, as the LLM can only access public test cases and the corresponding feedback. The reward function $r$ assigns a reward of 1 if the last time step code sample $a_T$ passes all test cases, where $T$ is the time horizon. The history $\tau_t = (o_0, a_1, o_1, \ldots, o_t)$ captures the whole trajectory of actions and observations up to the $t$-th time step. Finally, the policy $\pi$ outputs code revision based on the history for each time step, i.e., $a_{t+1} \sim \pi(\cdot | \tau_t)$.

**Local search.** Local search (Pirlot, 1996) is an important class of heuristic methods to solve computationally challenging optimization problems. Instead of systematically exploring the whole solution space, local search iteratively improves an incumbent solution by exploring its local neighborhood until a given termination condition is met. Local search naturally enjoys the anytime property (Zilberstein, 1996), in the sense that it can return a solution at anytime and the solution quality is monotonically non-decreasing over time by simply caching the best solution found so far.

## 3 Methodology

While existing improvement-based methods offer the advantage of iterative code revision through multi-turn interactions, they suffer from two critical limitations: (1) inefficient exploration due to complex revision workflows that consume excessive tokens, and (2) inaccurate reward functions that provide limited guidance for selecting promising candidates. To address these challenges, we introduce a lightweight and unified **Re**vision **Lo**cal Sear**c**h (ReLoc) framework, which leverages simple-yet-efficient decision rules to perform step-by-step code revision (cf. Figure 2). In Section 3.1, we elaborate the essentials and key algorithmic components of ReLoc. To address the limitations of prior ad-hoc evaluation heuristics, we further introduce a revision reward model trained to rank code candidates according to their revision distance in Section 3.2. Finally, we show the flexibility and expressiveness of our ReLoc framework by implementing two well-known local search algorithms (i.e., Hill Climbing and Genetic Algorithm) in Section 3.3.

### 3.1 Local Search Framework

As shown in Algorithm 1, our ReLoc framework consists of four algorithmic components that can be instantiated with different decision rules: (1) DRAFTCODE, which generates an initial code sample population $P_0$ based on the problem statement $x_i$ and public test cases $u_i$ of code generation task $\mathcal{T}_i$; (2) GENERATENEIGHBORHOOD, which constructs a set of neighborhood code samples $P_t$ by prompting the LLM $\pi$ to propose new code revisions based on the history and the execution feedback

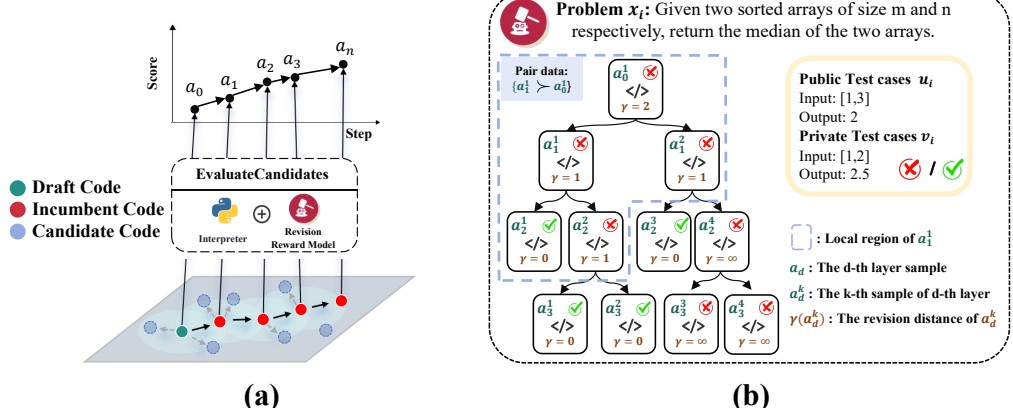

Figure 2: **(a) Overview of the ReLoc framework.** Starting from an initial draft code, ReLoc iteratively generates a neighborhood of candidate code samples around the current incumbent code. Each candidate is assessed by the EVALUATECANDIDATES, which utilizes an interpreter and our revision reward model. **(b) Training the Revision Reward Model.** A code tree is constructed by iteratively revising an initial incorrect code. Each code sample in the tree is labeled with its revision distance, i.e., the minimum number of revisions needed to reach a correct version. Pairwise comparisons in local region create preference data.

from the public test cases; (3) EVALUATECANDIDATES, which assigns each candidate in current neighborhood $P_t$ with a score measuring its quality; and (4) UPDATEINCUMBENT, which implements the move strategy by selecting the next incumbent solution $a_t$ from the neighborhood $P_t$. Finally, we maintain the best-so-far code sample $a^*$ with score $e^*$ to enforce the anytime property.

---

**Algorithm 1** ReLoc: Revision local search framework for code generation with LLMs

---

**Input:** Code generation task $\mathcal{T}_i$, LLM $\pi$, iteration limit $T$
**Output:** Code sample $a^*$
1: $P_0 \leftarrow$ generate initial code sample population with DRAFTCODE
2: $E_0 \leftarrow$ evaluate code samples $P_0$ with EVALUATECANDIDATES
3: $a_0 \leftarrow$ select a code sample from $P_0$ with UPDATEINCUMBENT, $a^* \leftarrow a_0$, $e^* \leftarrow E_0[a_0]$
4: **for** $t = 1, \ldots, T$ **do**
5:     $P_t \leftarrow$ generate neighborhood code samples with GENERATENEIGHBORHOOD
6:     $E_t \leftarrow$ evaluate code samples $P_t$ with EVALUATECANDIDATES
7:     $a_t \leftarrow$ select a code sample from $P_t$ with UPDATEINCUMBENT
8:     **if** $E_t[a_t] > e^*$ **then**
9:         $a^* \leftarrow a_t$, $e^* \leftarrow E_t[a_t]$
10: **return** $a^*$

---

### 3.2 Revision Reward Model

A key question in implementing ReLoc is how to evaluate the quality of the generated code candidates (cf. EVALUATECANDIDATES), which determines the search direction for each iteration. As we will show in Section 4.2, simple heuristics like pass rate on public test cases or LLM-based self-evaluation scores fail to effectively guide the search direction, since they often either collapse to binary signals or are prone to hallucinations. Outcome-based reward model (Shen and Zhang, 2024), on the other hand, solely focuses on the correctness of the code samples rather than how likely an incorrect candidate will be revised into a correct code in future steps, which is also not applicable to our scenario.

Instead of directly assessing the correctness, we train a specialized revision reward model to rank code samples according to their *revision distance*, i.e., the minimal number of revision steps required to transform it into the correct version. This way, in addition to prioritizing correct code samples, we also effectively differentiate incorrect ones and enable local search to focus on promising candidates with smaller revision distance, thus guaranteeing the overall search efficiency even when the correctness signals are uninformative (e.g., 0% pass rate on public test cases for all candidates).

To train our reward model, given a training task $\mathcal{T}_i$, we build a special code tree (cf. Figure 2(b)) where the root is an **incorrect** code $a_0^1$ sampled from the LLM policy. Then we incrementally expand the tree in a **breadth-first** fashion up to a depth limit $d_{\max}$. That is, for each incorrect code sample $a_{d-1}^j$ in the $(d-1)$-th layer, we prompt the LLM $\pi$ to generate $K$ code revisions for $d$-th layer:

$$a_d^{K(j-1)+k} \sim \pi\left(\cdot | REVISE\_PT\left(x_i, a_{d-1}^j, f(a_{d-1}^j; u_i)\right)\right) \qquad k \in \{1, \ldots, K\}, \qquad (1)$$

where $REVISE\_PT$ is the prompt template instructing the LLM $\pi$ to revise the code sample $a_{d-1}^j$ according to the problem statement $x_i$ and execution feedback $f(a_{d-1}^j; u_i)$ on public test cases, and the generated code revision $a_d^{K(j-1)+k}$ is then inserted to the tree as a child of $a_{d-1}^j$.

Once the code tree is built, we recursively label the revision distance of each code sample in the tree according to the following rule:

$$\gamma(a_d^j) = \begin{cases} 0, & a_d^j \text{ is correct;} \\ \infty, & a_d^j \text{ is incorrect } \wedge \ Ch(a_d^j) = \emptyset; \\ 1 + \min_{a \in Ch(a_d^j)} \gamma(a), & \text{otherwise,} \end{cases} \qquad (2)$$

where $Ch(a_d^j)$ is the children of $a_d^j$ in the code tree. Note that a code sample is considered correct if and only if it passes all public test cases $u_i$ and private test cases $v_i$. After that, we proceed to build win/loss pairs of code samples and train our reward model with Bradley–Terry framework (Bradley and Terry, 1952; Ouyang et al., 2022). Particularly, we confine the comparison within a small region around a code sample in the code tree to reflect the *locality* of the local search. Specifically, for code sample $a_d^j$, we consider the following neighborhood code samples:

$$\mathcal{N}(a_d^j) = \{Pa(a_d^j)\} \cup Ch(a_d^j) \cup Sib(a_d^j), \qquad (3)$$

where $Pa(a_d^j)$ is its parent and $Sib(a_d^j) = \{a | a \in Ch(Pa(a_d^j)) \wedge a \neq a_d^j\}$ is the siblings of $a_d^j$. Then for each code sample $a' \in \mathcal{N}(a_d^j)$ with $\gamma(a_d^j) < \gamma(a')$, we model the preference probability as

$$\mathbb{P}(a_d^j \succ a' | x_i) = \sigma\left(R_\phi(a_d^j | x_i) - R_\phi(a' | x_i)\right), \qquad (4)$$

where $\sigma(\cdot)$ is the sigmoid function and $R_\phi(a | x_i)$ represents the learned reward score for code samples $a$ given problem statement $x_i$. The learning objective of the reward model is to maximize the expected log-probability:

$$\max_\phi \mathbb{E}_{(x_i, a, a') \sim \mathcal{D}}[\log \mathbb{P}(a \succ a' | x_i)], \qquad (5)$$

where the pair dataset $\mathcal{D}$ is constructed by collecting the pairs of code samples in each code tree.

### 3.3 Case Study

To demonstrate the flexibility and expressiveness of our ReLoc framework, we now present two well-known local search algorithms, i.e., Hill Climbing (HC) (Russell and Norvig, 2016) and Genetic Algorithm (GA) (Mitchell, 1998) for improvement-based code generation with LLMs by instantiating each algorithmic component with specific decision rules.

**DRAFTCODE.**  It is widely acknowledged that the quality of the initial solution has a significant impact on the performance of local search (Lourenço et al., 2018). Therefore, to guarantee the quality of the initial code sample population $P_0$, we adopt a Plan-then-Generate paradigm by firstly prompting the LLM to enumerate $N$ diverse natural language plans that outline different algorithmic strategies. Then for each plan, we prompt the LLM to synthesize a corresponding code implementation, forming a candidate pool $P_0 = \{a_0^1, \ldots, a_0^N\}$.

**GENERATENEIGHBORHOOD.**  For each iteration $t$, we generate neighborhood code samples $P_t$ according to the following rules.

- **Hill Climbing.** Given the incumbent code sample $a_{t-1}$ and the feedback $f(a_{t-1}; u_i)$ from public test cases $u_i$, we generate the neighborhood code revisions by first prompting the

LLM to propose $K$ natural language revision strategies $Q_t = \{q_t^1, \ldots, q_t^K\}$ (e.g., "fix condition logic", "refactor loop"). Then for each strategy $q_t^k$, we prompt the LLM $\pi$ to generate a candidate code revision:

$$P_t = \{a_t^1, \ldots, a_t^K\}, \quad a_t^k \sim \pi\left(\cdot \mid HC\_PT\left(x_i, a_{t-1}, f(a_{t-1}; u_i), q_t^k\right)\right), \quad (6)$$

where $HC\_PT$ is the prompt template instructing the LLM to generate a code revision based on problem statement $x_i$, previous code $a_{t-1}$, execution feedback $f(a_{t-1}; u_i)$ and revision strategy $q_t^k$.

- **Genetic Algorithm.** For each iteration $t > 1$, we select two parent code samples $a, a'$ from the history $(a_0, \ldots, a_{t-1})$ according to their *fitness*, i.e., the scores evaluated by EVALUATECANDIDATES. Particularly, we select parent code samples from $P_0$ when $t = 1$. Then we prompt the LLM to generate $K$ new candidates:

$$P_t = \{a_t^1, \ldots, a_t^K\}, \quad a_t^k \sim \pi\left(\cdot \mid GA\_PT\left(x_i, a, f(a; u_i), a', f(a'; u_i)\right)\right), \quad (7)$$

where $GA\_PT$ is the prompt template[2] instructing the LLM to generate a code revision by combining the strengths or addressing the shared weaknesses of the parent code samples. To maintain diversity, we also implement an *aging* mechanism where each code sample is disqualified from being selected as a parent after it has been used in this role 3 times.

**EVALUATECANDIDATES.** To evaluate the candidate code samples $P_t$, we propose a synergistic approach that leverages both public test cases and the learned revision reward model. Let $P_{\text{pass}} = \{a \in P_t \mid f(a; u_i) = \texttt{pass}\}$ be the subset of candidates that pass all public test cases. Then, the evaluation score of candidate $a$ is defined as:

$$E_t[a] = \begin{cases} R_\phi(a|x_i), & \text{if } P_{\text{pass}} = \emptyset; \\ R_\phi(a|x_i), & \text{if } a \in P_{\text{pass}}; \\ -\infty, & \text{otherwise.} \end{cases} \quad (8)$$

That is, when no candidate passes all public tests, we fall back to using the reward model $R_\phi$ to score all candidates. If there are successful candidates, we score them using $R_\phi$, while assigning the rest with a score of $-\infty$. This way, we provide fine-grained preferences that guide the local search toward promising candidates by explicitly differentiating the candidates with the same correctness signal.

**UPDATEINCUMBENT.** Given the current code samples $P_t$ and the corresponding evaluation scores, UPDATEINCUMBENT aims to select a code sample as the incumbent solution for iteration $t$. Technically, decision rules like $\epsilon$-greedy, Boltzmann distribution (Landau and Lifshitz, 2013) or more complex simulated annealing acceptance rule (Delahaye et al., 2018) can be applied. Here we choose to greedily select the one with the maximum evaluation score for simplicity and efficiency, i.e., $a_t = \arg\max_{a \in P_t} E_t[a]$, where ties are broken alphabetically.

# 4 Experiments

In this section, we present extensive empirical evaluations to demonstrate our superiority across diverse code-related tasks. Our experiments aim to answer the following research questions:

- **RQ1:** How well do our local search methods perform on code-related tasks compared to state-of-the-art construction-based and improvement-based approaches?

- **RQ2:** Can our proposed revision reward model provide more effective guidance for local search than heuristic rewards or outcome-based reward model?

- **RQ3:** How does the performance of our local search methods scale with increasing token budgets at inference time?

- **RQ4:** What are the individual contributions of planning, revision strategies, and execution feedback to the overall performance of our local search methods?

---

[2] All prompt templates are provided in the Appendix.

Table 1: Pass@1 accuracy (%) of different methods on the LiveCodeBench and TACO benchmarks. All methods are evaluated under the same token budget (7K) to ensure fair comparison. Methods are categorized as construction-based (Con.) or improvement-based (Imp.), and further distinguished by reward functions: PRM (⚙️), self-evaluation (🤖), pass rate (🖥️), and revision reward model (🏅).

| Methods | Cat. | Rew. | LiveCodeBench | | TACO | |
|---|---|---|---|---|---|---|
| | | | Code gen | Code repair | Code gen | Code repair |
| RAP | Con. | ⚙️ | 22.9 | 21.2 | 5.4 | 4.1 |
| TOT | Con. | ⚙️ | 25.6 | 20.4 | 6.8 | 3.7 |
| Code Tree | Imp. | 🤖 | 27.7 | 24.1 | 8.2 | 5.6 |
| Reflexion | Imp. | 🤖 | 25.6 | 23.9 | 7.1 | 6.1 |
| Plan Search | Imp. | 🤖 | 32.7 | 31.3 | 11.2 | 8.2 |
| BoN | Imp. | 🖥️ | 30.2 | 30.5 | 10.8 | 6.3 |
| SFS | Imp. | 🖥️ | 32.1 | 27.3 | 10.5 | 8.1 |
| ORPS | Imp. | 🖥️ | 28.8 | 26.6 | 9.8 | 7.8 |
| ReLoc_HC (**Ours**) | Imp. | 🏅 | **38.4** | **33.4** | 13.3 | 9.7 |
| ReLoc_GA (**Ours**) | Imp. | 🏅 | 35.7 | 29.9 | **15.3** | **11.5** |

## 4.1 Experimental Setup

**Benchmarks.** We evaluate our methods on two benchmarks: **LiveCodeBench** (Jain et al., 2024), using 511 problems from May 2023–May 2024 for training and 268 problems from Jul 2024–Jan 2025 for testing; and **TACO** (Li et al., 2023), which aggregates problems from CodeContests, APPS (Hendrycks et al., 2021), and other sources. From TACO's original 25,443 training and 1,000 test problems, we randomly sample 4,000 and 200 respectively due to computational constraints. Both datasets provide 2–3 public test cases per problem. For code repair, we construct a buggy-code dataset by sampling incorrect solutions from diverse models (Qwen2.5-7B/32B/70B, GPT-4o) to ensure a range of error types and complexities.

**Implementation details.** Our base model throughout the experiments is Qwen2.5-32B-Instruct. During training, we use the training splits of Live-CodeBench and TACO to construct code revision pairs as described in Section 3.2, where we expand code trees via breadth-first search up to a maximum depth of $d_{max} = 5$, with $K = 3$ revisions per node. These pairs are used to train a specialized revision reward model based on Qwen2.5-7B-Instruct, following the reward modeling procedure from (von Werra et al., 2020). The distribution of pairwise comparisons is shown in Figure 3. For inference, we adopt decoding settings consistent with (Jain et al., 2024), using a temperature of 0.2 and top-$p$ of 0.95. To ensure fair comparison across all methods, we fix the token budget to 7K tokens per problem. Additional implementation and training details are provided in the Appendix.

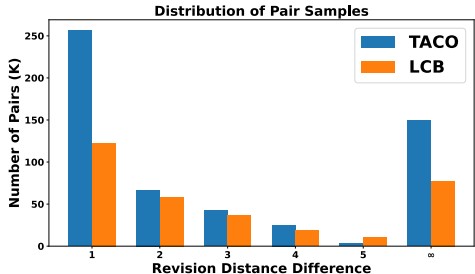

Figure 3: Distribution of 872K training pairs by revision distance difference ($\gamma(\text{loss}) - \gamma(\text{win})$).

**Baselines.** We compare our methods with the following approaches. (1) **Construction-based** approaches like ToT (Yao et al., 2023) and RAP (Hao et al., 2023), which build solutions via search (BFS or MCTS); (2) **Improvement-based methods with self-evaluation**, which leverage internal assessment mechanisms to guide search, such as CodeTree (Li et al., 2024a) and Reflexion (Shinn et al., 2023), which use multi-agent evaluation or test case generation with reflective learning; and Plan-and-Search (Wang et al., 2024c), which prompts LLMs to search among candidate solution plans expressed in natural language. (3) **Improvement-based methods with pass rate**, which utilize performance on public test cases to provide reward signals, including Best-of-N (BoN) (Cobbe et al.,

2021), which randomly samples $N$ solutions and filters according to the pass rate; SFS (Light et al., 2024), which employs MCTS to revise code using pass rate as the reward function; and OPRS (Yu et al., 2024), which combines self-scoring and pass rate to guide beam search.

## 4.2 Empirical Results

**Performance comparison.** We systematically compare our local search methods against state-of-the-art baselines and present the results in Table 1. It can be concluded that ReLoc consistently outperforms all baseline methods across both LiveCodeBench and TACO benchmarks on the Pass@1 metric. Specifically, ReLoc_HC achieves the best performance of **38.4%** and **33.4%** on Live-CodeBench, while ReLoc_GA reaches **15.3%** and **11.5%** on TACO. It is interesting to find that ReLoc_HC outperforms ReLoc_GA on the easier LiveCodeBench benchmark, while the reverse is observed on the more challenging TACO tasks. This phenomenon highlights the distinct merits of each algorithm: ReLoc_HC is particularly effective for straightforward tasks where the underlying solution is relatively obvious and the primary challenge lies in fixing minor bugs or syntax errors, while ReLoc_GA excels in more complex scenarios by strategically integrating the advantages from parent code samples to discover non-trivial solutions for conceptually challenging tasks in TACO.

Compared to construction-based methods like ToT and RAP, ReLoc exhibits significantly superior performance under the same computational budget, thanks to the anytime property of local search algorithms. On the other hand, improvement-based methods with self-evaluation often are prone to hallucinations, leading to unreliable assessments of code quality. While such methods can be somewhat effective for guiding high-level search, as evidenced by Plan Search, they cannot fully leverage the detailed execution feedback. Unlike Plan Search that explores high-level strategies in an abstract plan space, ReLoc implements fine-grained planning and code-level revision, achieving an average improvement of **4.9%** over Plan Search on Code gen tasks.

Finally, the improvement-based techniques using pass rates from public test cases struggle with sparse or binary reward signals, which also offer insufficient guidance for search. SFS-based MCTS methods require extensive exploration with high computational costs, while ORPS employing beam search suffers from low exploration efficiency, unable to select the most promising code candidates. ReLoc adeptly navigates these challenges through lightweight decision rules, guided by a revision reward model, achieving an average improvement of **5.6%** over pass rate-based methods across different tasks.

**Effectiveness of revision reward model.** To evaluate the effectiveness of our revision reward model when guiding local search, we conduct controlled experiments on LiveCodeBench and TACO on ReLoc_HC with different reward functions. Specifically, we compare revision reward model against five baselines: public test case pass rate, LLM-generated test case pass rate, LLM self-evaluation, Skywork-27B reward model (Liu et al., 2024), and ORM-7B, i.e., an outcome-based reward model trained with the same architecture as the revision reward model (`Qwen2.5-7B`).

As shown in Table 2, pass rate and self-evaluation heuristics offer weak guidance. That is not surprising because execution-based scores are often coarse or binary, while self-evaluated scores often suffer from hallucinations, leading to unreliable rankings. In contrast, methods using a reward model perform better, with our revision reward model outperforming ORM by **35.9%→38.4%** and **9.7%→13.3%** on LiveCodeBench and TACO. We attribute this to the ability of the revision reward model to differentiate incorrect candidates based on their likelihood of future correction, which is a property the ORM lacks due to its exclusive focus on code correctness.

**Inference-time scaling law.** To further evaluate the scaling performance of ReLoc_HC with increasing computational resources, we vary the token budget from 1K to 15K. We compare our method against three representative baselines: BoN, ORPS, and ToT. Figure 4 illustrates the scaling behavior of different methods on the LiveCodeBench and TACO benchmarks. Notably, as the token budget increases, ReLoc_HC demonstrates a faster improvement in terms of Pass@1, highlighting the benefit of guided local search and our learned revision reward model. This is particularly evident in the LiveCodeBench, where ReLoc_HC reaches over **40%** Pass@1 under a 15K token budget, outperforming the BoN with the same budget by a significant margin.

Table 2: Pass@1 accuracy of ReLoc_HC with different reward functions under a 7K token budget. Our revision reward model achieves the highest Pass@1 on both LiveCodeBench and TACO, demonstrating strong inference-time performance without relying on test case generation or self-evaluation.

| Reward Function | Gen Test Case | Self Score | Reward Model | LiveCodeBench | TACO |
|---|---|---|---|---|---|
| Pass Rate | ✗ | ✗ | ✗ | 33.5 | 10.3 |
|    w/ Gen Test case | ✓ | ✗ | ✗ | 29.0 | 8.4 |
| Self Evaluation | ✓ | ✓ | ✗ | 29.3 | 9.2 |
| Skywork-27B | ✗ | ✗ | ✓ | 31.9 | 9.4 |
| ORM-7B | ✗ | ✗ | ✓ | 35.9 | 9.7 |
| Revision Reward Model (**Ours**) | ✗ | ✗ | ✓ | **38.4** | **13.3** |

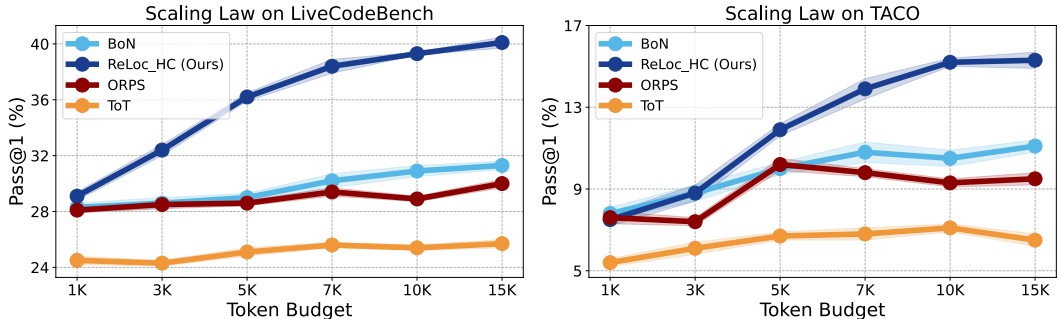

Figure 4: **Scaling Law.** Pass@1 accuracy of ReLoc_HC (**Ours**) compared to baselines (BoN, ORPS, ToT) on LiveCodeBench and TACO benchmarks as token budget increases from 1K to 15K.

Table 3: **Ablation study of ReLoc.** We evaluate the impact of core components in both **ReLoc_HC** and **ReLoc_GA**. Removing natural language plans, revision strategies, or execution feedback.

| Method | LiveCodeBench | | TACO | |
|---|---|---|---|---|
| | **Pass@1 (%)** | **Tokens (1K)** | **Pass@1 (%)** | **Tokens (1K)** |
| ReLoc_HC | 38.4 | 7.1 | 13.3 | 7.6 |
|    w/o Natural Language Plans | 36.9 | 6.5 | 13.7 | 7.1 |
|    w/o Revision Strategies | 35.9 | 4.7 | 11.7 | 5.5 |
|    w/o Execution Feedback | 34.8 | 7.5 | 11.4 | 7.7 |
| ReLoc_GA | 35.7 | 6.8 | 15.3 | 7.7 |
|    w/o Natural Language Plans | 34.3 | 4.9 | 12.9 | 6.6 |
|    w/o Execution Feedback | 34.6 | 5.8 | 13.5 | 6.8 |

**Ablation study.** We conduct an ablation study on both **ReLoc_HC** and **ReLoc_GA** to assess the importance of each design choice. As shown in Table 3, removing natural language plans during initialization and replacing them with randomly sampled code reduces performance. Notably, eliminating revision strategies significantly reduces the number of generated tokens (e.g., from 7.1K to 4.7K in ReLoc_HC). However, this also limits the diversity of candidate code samples explored during the search, ultimately resulting in inferior performance. Furthermore, execution feedback plays a particularly crucial role, as it enables precise and targeted revision in each iteration, improving the overall Pass@1 accuracy by 2.1%.

# 5 Conclusion

In this work, we present ReLoc, a lightweight and unified local search framework for improvement-based code generation with LLMs. Unlike computationally expensive construction-based inference-time scaling methods like ToT and MCTS, ReLoc finds high-quality solutions and enjoys the anytime property by exploring a series of local revisions of an established code sample. Besides, compared to the existing improvement-based methods, ReLoc leverages simple yet effective decision rules to navigate the search space. Furthermore, a specialized revision reward model effectively differentiates code samples based on the potential of each code sample being corrected in future steps, which provides fine-grained preferences when the correctness signal is uninformative. Finally, we show the flexibility and expressiveness of ReLoc by developing two well-known local search algorithms, i.e., Hill Climbing and Genetic Algorithm. Extensive experiments on benchmarks like LiveCodeBench and TACO validate the effectiveness of ReLoc in significantly improving code generation performance while reducing computational cost.

## Acknowledgments

This research is supported by the RIE2025 Industry Alignment Fund – Industry Collaboration Projects (IAF-ICP) (Award I2301E0026), administered by A*STAR, as well as supported by Alibaba Group and NTU Singapore through Alibaba-NTU Global e-Sustainability CorpLab (ANGEL).

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

# A  Related Work

**Code Generation with Large Language Models.** Recent advances in large language models (LLMs) have significantly boosted code generation by leveraging pretraining on large-scale code corpora (Lu et al., 2021; Christopoulou et al., 2022; Guo et al., 2024; Hui et al., 2024). At inference time, two main strategies have emerged to further enhance performance. Construction-based methods generate solutions step-by-step, often guided by value models (Yao et al., 2023; Wang et al., 2024b), process-based reward models (Lightman et al., 2023), or planning techniques like Monte Carlo Tree Search (Hao et al., 2023; Zhou et al., 2023). In contrast, improvement-based approaches iteratively refine full code drafts using multi-turn updates, agentic workflows (Wang et al., 2024a; Zhong et al., 2024; Zhang et al., 2024b), and test-time feedback, sometimes with multi-agent collaboration (Li et al., 2024a; Zan et al., 2024). While effective, many of these methods are resource-intensive and complex. Our work introduces a lightweight local search framework that streamlines key components into an efficient and scalable loop.

**Reward Models.** Reward models play a central role in RLHF (Ouyang et al., 2022), providing learning signals for policy optimization. To reduce dependence on human-labeled data, RLAIF (Lee et al., 2023) proposed an automated reward data pipeline. More recently, reward models have been extended to reasoning tasks. Math-Shepherd (Wang et al., 2023) and others (Zhang et al., 2025) trained process-based reward models to guide inference-time strategies, while outcome-based models have supported tree search (Jiang et al., 2024). Generative Reward Models (GRMs) (Mahan et al., 2024; McAleese et al., 2024; Zhang et al., 2024c; Chen et al., 2025) further leverage CoT-based self-critique for scoring outputs. Distinct from these paradigms, we propose a reward model trained to estimate *revision distance*, capturing the minimal steps needed to reach a correct solution. This enables more efficient candidate evaluation and improves the effectiveness of iterative code refinement.

# B  Experimental Setup.

## B.1  Revision Reward Model

This section outlines the hyperparameters and settings used during the training phase of the revision reward model. We trained the revision reward model on `Qwen2.5-7B-Instruct` using the TRL library (von Werra et al., 2020) with DeepSpeed ZeRO Stage 2 on 4 NVIDIA H100 GPUs. The training was conducted for one epoch on a combined dataset of LiveCodeBench and TACO. Table 4 details the training configuration.

Table 4: Revision reward model training Configuration

| Parameter | Value |
| --- | --- |
| Mixed Precision | bf16 |
| Batch Size per Device | 8 |
| Number of Epochs | 1 |
| Gradient Checkpointing | True |
| Learning Rate | 5.0e-6 |
| Logging Steps | 25 |
| Evaluation Strategy | Steps |
| Evaluation Interval | Every 500 steps |
| Save Interval | Every 3000 steps |
| Max Sequence Length | 2048 |
| Push to Hub | False |
| Optimizer | paged_adamw_32bit |
| Warmup Ratio | 0.05 |
| Learning Rate Scheduler | Cosine |
| Number of GPUs | 4 × NVIDIA H100 |

### B.2 Local Search Hyperparameters

We use `Qwen2.5-32B-Instruct` as the inference model throughout all experiments, with a decoding temperature of 0.2 and a top-$p$ value of 0.95. All algorithms are run under a fixed token budget of 7,000 tokens per task.

For **Hill Climbing (HC)**, we initialize with 5 draft codes and expand 3 neighbors for each candidate during each improvement iteration.

For the **Genetic Algorithm (GA)**, we similarly maintain a population of 5 draft codes. In each iteration, we select 2 codes from the candidate pool as parent codes, with each code allowed to be selected as a parent up to 3 times.

## C  Additional Evaluation on GPT-4o

To further validate the effectiveness of RELOC, we conduct experiments on the closed-source model `gpt-4o-2024-1120`. Specifically, we evaluate RELOC and several baselines, including the state-of-the-art *Plan Search* algorithm and *Best of N* sampling, on the LIVECODEBENCH benchmark. For RELOC, we employ the revision reward model trained as described in Section B.1 to guide the search process.

In our evaluation, the revision reward model guiding RELOC's local search was trained entirely on data sampled from the open-source `Qwen2.5-32B-Instruct` model, a setting that differs in both distribution and model family from the target inference model, GPT-4o. Remarkably, as shown in Table 5, RELOC achieves the highest Pass@1 score (44.2%) while consuming only 8.7K tokens on average—representing a 52% reduction in token usage compared to *Plan Search*. This underscores the efficiency of RELOC's local search mechanism.

More importantly, these results demonstrate that the revision reward model, trained on Qwen-generated code trajectories, generalizes robustly to guide search on GPT-4o. This transferability is non-trivial: GPT-4o may exhibit different stylistic tendencies, error patterns, and semantic representations compared to Qwen2.5-32B. Yet, the reward model still provides reliable signals for ranking candidate revisions, suggesting that it captures model-agnostic features of code quality—such as syntactic closeness to correct solutions, functional coherence, and local editability.

Such robustness to distributional shifts suggests broader applicability of our approach. It indicates that RELOC, and particularly its reward model component, can serve as a plug-and-play module to improve inference-time performance across diverse LLMs, without the need for costly re-annotation or model-specific retraining. This is particularly valuable for deployment in scenarios involving closed-source or frequently updated models, where direct supervision signals or fine-tuning access are unavailable.

Table 5: Performance comparison on GPT-4o (`gpt-4o-2024-1120`).

| Methods | Rew. | Pass@1 (%) | Tokens (1K) |
|---|---|---|---|
| Plan Search | self-evaluation | 42.7 | 16.7 |
| BoN | Pass Rate | 41.8 | 12.2 |
| ReLoc_HC (**Ours**) | Revision Reward Model | **44.2** | **8.7** |

# D Prompts

Table 6: Write diverse observations based on the problem to guide the generation of draft code.

| Attribute | Description |
|---|---|
| DRAFTCODE | You are an expert Python programmer. You will be given a competitive programming question (problem specification). You will return 5 useful, non-obvious, and correct observations about the problem, like hints to solve the problem. You will NOT return any code. Be as creative as possible, going beyond what you think is intuitively correct. For each observation should be between tag [OBSERVATION] and [/OBSERVATION]. |
| Outputs | **[OBSERVATION]**
Observation 1: A permutation can be fixed with at most one swap if and only if it has at most 2 inversions.
**[/OBSERVATION]**

**[OBSERVATION]**
Observation 2: Only 3-cycles (like "bca" and "cab") cannot be solved with a single swap operation.
**[/OBSERVATION]**

... |

Table 7: ReLoc_HC generates diverse modification strategies for incumbent code

| Attribute | Description |
|---|---|
| HC_NEIGHBOR | You are an expert in debugging Python code. You will be provided with a code snippet that requires debugging, along with a revision history for reference. Your task is to begin by writing a brief textual explanation of the current code—summarize its intended behavior and any evident issues between [explanation] and [/explanation]. Next, propose three refinement directions that could help improve the code. Please put each direction between tag [direction] and [/direction]. Important: Do not include any Python code in your response—only the explanation and the three refinement directions. |
| Outputs | **[explanation]**
The feedback indicates that the main issue is the function returns only the first element of min-k and max-k, instead of the entire lists.
**[/explanation]**

**[direction]**
Modify the return statement to return the full min-k and max-k lists.
**[/direction]**

**[direction]**
Include input validation to ensure K is a non-negative integer.
**[/direction]**

... |

Table 8: ReLoc_HC revises code according to different strategies

| Attribute | Description |
|---|---|
| HC_NEIGHBOR | You are a helpful programming assistant and an expert in Python. The user has written code that contains errors. You will be provided with a Python programming problem, the user's code, revision history, an explanation, and directions for refinement. Your task is to debug and revise the code to correctly solve the problem. Generate a corrected version of the complete program, incorporating the explanation, refinement directions, and revision history. Output only one corrected program enclosed within a single pair of code delimiters. Do not include any additional commentary or text. |
| Outputs | <pre>def minimumBoxes(apple_packs, box_capacities):
    for box_capacity in box_capacities:
        current_capacity += box_capacity
        boxes_used += 1
        if current_capacity >= total_apples:
            return boxes_used
    return boxes_used

apple_packs = [1, 3, 2]
box_capacities = [4, 3, 1, 5, 2]
print(minimumBoxes(apple_packs, box_capacities))

apple_packs = [5, 5, 5]
box_capacities = [2, 4, 2, 7]
print(minimumBoxes(apple_packs, box_capacities))</pre> |

Table 9: ReLoc_GA directly prompts LLM to generate neighborhood by referencing parent code

| Attribute | Description |
|---|---|
| GA_NEIGHBOR | You are a helpful programming assistant and an expert in Python. The user has written code that contains errors. You will be provided with a Python programming problem and the user's code intended to solve it. Your task is to refer to the input code and revise it to correctly solve the problem. Generate a corrected version of the complete program. Output only one corrected program enclosed within a single pair of code delimiters. Do not include any additional commentary or text. |
| Outputs | <pre>def minimumBoxes(apple_packs, box_capacities):
    for box_capacity in box_capacities:
        current_capacity += box_capacity
        boxes_used += 1
        if current_capacity >= total_apples:
            return boxes_used
    return boxes_used

apple_packs = [1, 3, 2]
box_capacities = [4, 3, 1, 5, 2]
print(minimumBoxes(apple_packs, box_capacities))

apple_packs = [5, 5, 5]
box_capacities = [2, 4, 2, 7]
print(minimumBoxes(apple_packs, box_capacities))</pre> |

# E  Step by Step Revisions

Below, we demonstrate how ReLoc guides `Qwen2.5-32B-Instruct` step by step to revise completely incorrect draft code into correct code.

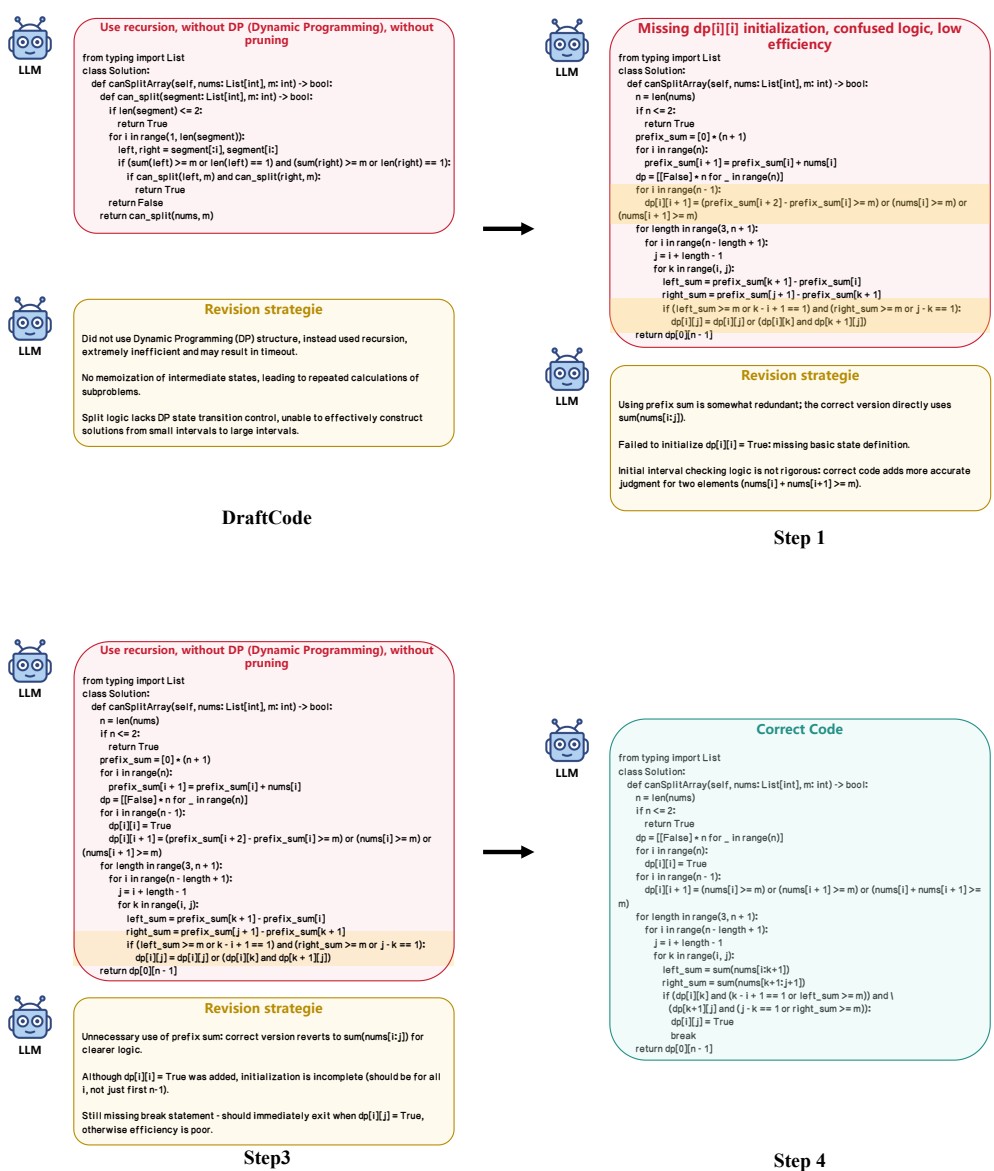

Figure 5: ReLoc step-by-step revise incorrect code

