# OpenReview forum: "Let's Revise Step-by-Step: A Unified Local Search Framework for Code Generation with LLMs"
_NeurIPS.cc/2025/Conference — NeurIPS 2025 poster_

### Official Review · Reviewer_ZiY2 · 2025-07-02

**Clarity:** 2
**Significance:** 3
**Originality:** 3
**Rating:** 3
**Confidence:** 3

**Summary:**

This paper proposes ReLoc, a framework for improving LLM-based code generation through iterative revision. The authors frame the problem as a local search, where an initial code draft is progressively refined. The framework can be instantiated with different local search algorithms, with the paper focusing on Hill Climbing (HC) and a Genetic Algorithm (GA). A key contribution is a specialized "revision reward model" trained to estimate the "revision distance"—the number of steps needed to fix a piece of code. This model is intended to provide a more fine-grained evaluation signal than simple pass/fail rates on test cases. The authors conduct experiments on the LiveCodeBench and TACO benchmarks, claiming that their approach outperforms existing construction-based and improvement-based methods in both performance and efficiency

**Questions:**

- The data generation for the revision reward model, which involves building large code trees via breadth-first search, seems extremely expensive. Could you quantify the computational cost (e.g., total tokens, GPU hours) required to create the training dataset for this model? How does this pre-computation cost weigh against the claimed inference-time efficiency gains?

- The abstract claims a "37% reduction in token consumption," citing Figure 4. Could you please provide a clear, step-by-step calculation showing how this number was derived? Which specific methods, tasks, and performance levels are being compared to arrive at this figure?

- The "crossover" operation for the Genetic Algorithm is critical but is described abstractly as prompting an LLM to combine strengths. Could you please provide the exact prompt template (GA_PT) used? Furthermore, what evidence do you have that this approach produces meaningful and diverse offspring, rather than simply averaging or slightly modifying one of the parents?

- The ablation study shows that removing explicit revision strategies causes a relatively small drop in performance. This seems to contradict the paper's emphasis on guided, step-by-step revision. Could you elaborate on why this component has such a limited impact and whether this undermines the core motivation for the framework?

**Ethical Concerns:**

["NO or VERY MINOR ethics concerns only"]

**Final Justification:**

My major concerns have now been addressed, and I have adjusted my overall score for the paper accordingly.

**Limitations:**

Yes.

**Quality:**

2

**Strengths And Weaknesses:**

**Strengths**:

The paper addresses the important and timely problem of scaling LLM inference for complex code generation tasks. The core idea of using a more nuanced reward signal that goes beyond binary pass/fail feedback is a reasonable direction for research. The proposed framework is also flexible in principle, capable of incorporating various search heuristics.

**Weaknesses**:

- The central proposal is to apply classic local search heuristics (Hill Climbing, Genetic Algorithms) to the output of an LLM. These are standard, well-understood algorithms from the field of optimization. While their application to LLM-based code generation is relevant, it represents an incremental step rather than a novel breakthrough. The paper frames this as a "unified framework," but this feels like an overstatement for what is essentially the implementation of two common search methods. The novelty, therefore, rests almost entirely on the "revision reward model."

- While the method shows performance gains, the significance of these gains is debatable. For instance, on the code generation task in LiveCodeBench, ReLoc_HC (33.4%) is only marginally better than the Plan Search baseline (32.7%). The reliance on a complex, separately trained reward model, which itself requires a massive data generation effort, seems like a very high price to pay for such modest improvements. The work adds significant complexity to the overall pipeline for gains that are not consistently transformative.

- The ablation study in Table 3 is meant to show the importance of each component, but it inadvertently raises questions. Removing "Revision Strategies" from the Hill Climbing variant results in only a minor performance drop (e.g., 38.4% -> 35.9% on LiveCodeBench). This suggests that the structured, multi-step revision planning, a key part of the described method, has a surprisingly small impact, weakening the overall narrative about the importance of fine-grained, planned revisions.

---

> ### Author Rebuttal · Authors · 2025-07-31
>
> We sincerely appreciate your thoughtful and constructive review. Thank you for recognizing the motivation and promise of our approach, especially the value of using fine-grained reward signals. Below, we address your remaining questions and concerns in detail.
>
> > W1: The use of classical local search algorithms (Hill Climbing, Genetic Algorithms) in this work is viewed as incremental rather than novel.
>
> Thank you for raising this important point. We agree that hill climbing and genetic algorithms are classic search strategies and not novel on their own. However, our contribution lies not in simply reusing these algorithms, but in **adapting and integrating them into a principled and effective local search framework tailored for LLM-based code generation**, which faces unique challenges compared to classical optimization problems. We would like to clarify the sources of novelty in our work:
>
> 1. **Adapting Classical Local Search for LLM-Based Code Generation.**
>
>    * **While hill climbing and genetic algorithms are classical methods, applying them effectively to LLM-based code generation is far from straightforward.** Traditional local search relies on well-defined, dense objective functions, whereas in code generation, feedback is often sparse, binary, and noisy. Moreover, candidate programs are structured and long-form, making it non-trivial to sample meaningful local neighborhoods under tight token budgets.
>
>     * **To overcome these challenges, we build on the core idea of iterative refinement.** Based on this perspective, we propose a modular framework tailored for LLMs, which decomposes the process into four components: initialization, neighborhood generation, candidate evaluation, and incumbent update.
>
>     * **Each module is designed to support flexibility and task-specific adaptation.** For instance, initialization can use plan-based prompting or pure sampling depending on the scenario. Neighborhood generation can leverage execution signals for repair tasks. Classic algorithms like hill climbing and genetic algorithms are adapted to operate over these components in a way that respects the nature of LLMs and the discrete, structured nature of code.
>
> 2. **Revision Reward Model Tailored for Local Search.**
>
>     We agree that the reward model is a core component of our approach. However, it is not an isolated module that can be separated from the framework. It is designed specifically to address a **fundamental challenge of evaluating candidates within neighborhoods during the local search process.** Inspired by the local search principle of leveraging structural locality, our revision reward model uses revision distance to provide fine-grained, relative feedback based on relationships among parent, child, and sibling code candidates.
>
> > W2: Marginal gains (32.7% to 33.4%) may not justify the added complexity of training the revision reward model.
>
> We would like to clarify that ReLoc_HC improves performance from **32.7% to 38.4%** on the code generation task, compared to Plan Search, and also achieves better results on the code repair task. These gains demonstrate that even with a lightweight local search procedure, our approach can effectively leverage the revision reward model to guide the search process.
>
> | Method| Code Generation | Code Repair |
> |-|-|-|
> | Plan Search          | 32.7            | 31.3        |
> | **ReLoc\_HC (Ours)** | **38.4**        | **33.4**    |
> | **ReLoc\_GA (Ours)** | 35.7            | 29.9        |
>
> **Training the revision reward model involves a one-time cost, but it offers broad and lasting utility.** Once trained, the model can be reused across different tasks and model families without retraining. To demonstrate this, we apply the reward model trained solely on data from `Qwen2.5-32B-Instruct` to a stronger frontier model, `GPT-4o`, and observe continued performance gains with even better token efficiency, as shown below:
>
> | Methods| Reward Signal| Pass\@1 (%) | Tokens (1K) |
> |-|-|-|-|
> |Plan Search| self-evaluation|42.7|16.7|
> |BoN| Pass Rate | 41.8| 12.2 |
> |**ReLoc\_HC (Ours)**| Revision Reward Model|**44.2**| **8.7** |
>
> > W3 & Q4: The importance of the revision strategy.
>
> Thank you for identifying this area where additional clarification is needed.
>
> In the ReLoc, the `Revision Strategies` component refers to prompting the LLM to first generate **natural language descriptions** of revision plans (e.g., “Failed to initialize, missing basic state definition”), which are then used to guide code edits during the neighborhood generation. This is one specific sampling strategy within the neighborhood generation module.
>
> As you correctly observed, when we ablate this component and instead directly prompt the LLM to revise code without revision descriptions, the performance drops only modestly (e.g., 38.4% → 35.9%). This result highlights **ReLoc performs well even without this natural language planning**.
>
> > Q1: The computational cost of generating the training data. How does this pre-computation cost weigh against the claimed inference-time efficiency gains?
>
> Thank you for your insightful question regarding the pre-computation cost for training the revision reward model. While data sampling introduces some overhead, **the use of BFS makes it efficient, and the reward model is trained only once**. Below, we address the computational cost from two perspectives:
>
> 1. **Efficient Data Sampling via BFS and Locality-Based Comparison.**
>
>     * We limit the maximum depth to `d_max=5` and the branching factor to `K = 3` to avoid combinatorial explosion. In practice, due to duplicate or correct code samples during sampling, each code tree contains **only \~11 unique nodes on average**.
>
>     * More importantly, our comparison strategy leverages locality, comparing each node to its parent, children, and siblings. As a result, **each node participates in 2.4 comparisons on average**, making efficient use of every sampled code snippet.
>
>     * To further scale this process, we build a highly parallelized sampling system using vLLM, running on 4×H100 GPUs, completing data collection in **2.3 days**, generating **178M tokens**. This cost is entirely manageable in modern training pipelines and requires no human annotation.
>
> 2. **Robust Training with Small Subsets of Data.**
>
>     We perform additional experiments to test the sample efficiency of the revision reward model. As shown below, **the model maintains strong performance even with significantly reduced training data**:
>
>     | Downsampling Strategy| LiveCodeBench | TACO|
>     |-|-|-|
>     | Random Quarter (218K pairs) | 38.1| 13.2|
>     | Random Half (436K pairs)| 38.3| 13.1|
>     | Full-scale Training (872K)|38.4| 13.3 |
>
> > Q2: Clarify how the 37% token reduction is calculated
>
> We appreciate the opportunity to clarify the source and calculation of the "37% reduction in token consumption".
>
> While this figure was referenced alongside Figure 4, it is in fact derived from a separate comparison **between ReLoc and Plan Search** under unconstrained token settings, as shown in the table below. This analysis evaluates both performance and token usage on the code generation tasks in LiveCodeBench and TACO:
>
> || **LiveCodeBench** || **TACO**||
> |-|-|-|-|-|
> | **Method**| **Pass\@1 (%)**   | **Tokens (1K)** | **Pass\@1 (%)** | **Tokens (1K)** |
> | Plan Search | 33.8 |11.6| 11.5| 11.6  |
> | ReLoc\_HC (Ours) | **38.4**  | 7.1 | 13.3 | 7.6 |
> | ReLoc\_GA (Ours) | 35.7 | 6.8| **15.3** | 7.7|
>
> Averaging across tasks, **ReLoc uses 7.3K tokens** compared to **11.6K** for Plan Search, resulting in a 37% reduction in token consumption, while also achieving better overall accuracy.
>
> > Q3: Clarify GA crossover prompt and validate offspring diversity and effectiveness.
>
> Thank you for pointing this out. In the ReLoc framework, assessing the diversity of offspring is indeed crucial, especially for population-based algorithms such as Genetic Algorithm (GA). We have included the exact prompt template used for  neighborhood generation in **Appendix Section D, Table 6**.
>
> To evaluate the diversity of the generated offspring, we adopt two evaluation strategies:
>
> * **CodeBLEU analysis** \[1]:
>   We compute CodeBLEU between offspring and parents to assess non-trivial changes. A lower scores indicate meaningful edits beyond simple averaging or minor tweaks.
> * **BERT-based diversity measure** \[2]:
>   Following prior work, we measure diversity by averaging BERT cosine similarity across candidates. Lower scores reflect greater diversity.
>
> We also include Reflexion, a well-known multi-step refinement method, as a baseline for comparison. The results below are reported on the LiveCodeBench benchmark.
>
> | **Method**| **CodeBLEU ↓** | **Syntax Match ↓** | **Dataflow Match ↓** | **n-gram Match ↓** | **BERT Sim ↓** |
> |-|-|-|-|-|-|
> | Reflexion | 0.8722| 0.9017| 0.8748| 0.8498| 0.9987|
> | ReLoc\_GA (Ours)|**0.7364**| **0.7999**| **0.7671**| **0.6686** | **0.9948** |
>
> **ReLoc\_GA generates offspring that are more structurally and semantically different from their parents**, and promotes greater population diversity, demonstrating effective exploration behavior.
>
> \[1] Ren, Shuo, et al. "Codebleu: a method for automatic evaluation of code synthesis." arXiv preprint arXiv:2009.10297 (2020).
> \[2] Light, Jonathan, et al. "Scattered forest search: Smarter code space exploration with llms." arXiv preprint arXiv:2411.05010 (2024).
>
> Thank you again for all your constructive feedback. We will include these clarifications, especially the revision strategies, and updates in our revision.

---

> > ### Comment · Reviewer_ZiY2 · 2025-08-04
> >
> > Thank you for your careful and detailed response. My major concerns have now been addressed, and I have adjusted my overall score for the paper accordingly.

---

> > > ### Author Response · Authors · 2025-08-05
> > >
> > > We appreciate your comments and positive response. Thank you for all the time and efforts in reviewing this paper.

---

### Official Review · Reviewer_n8zP · 2025-07-03

**Clarity:** 3
**Significance:** 3
**Originality:** 2
**Rating:** 5
**Confidence:** 4

**Summary:**

This paper introduces ReLoc, a new framework for search based code generation. The authors identify that existing methods for code generation using LLMs, such as construction based tree search and improvement based methods, have significant drawbacks. Construction based methods are inefficient and not scalable, while improvement based methods often use uninformative reward signals and inefficient search strategies. To address these issues, ReLoc uses a step by step revision process to improve upon an initial code draft. The framework has four main components:drafting an initial code version, neighborhood code generation, candidate evaluation using reward model, and incumbent code updating. The authors developed a special revision reward model that is used to evaluate the quality of the code candidates. This model is trained to prefer code that is closer to a correct solution, which helps to guide the search. The authors show that ReLoc can be used to implement different local search algorithms like Hill Climbing and Genetic Algorithm. The experimental results show that ReLoc outperforms other methods on two different code generation benchmarks, LiveCodeBench and TACO.

**Questions:**

- Can you provide a comparison of algorithms using more capable frontier models? In particular, use GPT, Sonnet, or Gemini instead of Qwen 32B as the generation model. This experiment will tell us more about whether one search framework is more or less effective with better models.
- Paper is using Plan then Generate paradigm to create the initial code sample population. How important is the quality of the initial code to the final performance of ReLoc? Have you experimented with other methods for generating the initial code, such as simply sampling from the LLM and using the reward model to select an initial candidate?
- What is the accuracy of reward model in identifying a correct code sample given a correct and incorrect solution pair on unseen samples?
- How did you decide 7k token budget for experiments? From Fig 4, it seems ReLoc keep improving with larger context budget compare to other baselines.

**Ethical Concerns:**

["NO or VERY MINOR ethics concerns only"]

**Final Justification:**

After reading rebuttal and fellow reviewers feedback, I have decided to keep my positive review as I believe the proposed edit distance based reward design is a good contribution to the community.

**Limitations:**

yes

**Quality:**

3

**Strengths And Weaknesses:**

Strengths:

- The proposed ReLoc framework is well designed and the different components are clearly explained. The revision reward model is a clever way to guide the local search and overcome the limitations of using simple pass rates or self evaluation scores.
- The experimental evaluation is extensive and the results are promising. The authors compare their method with a wide range of baselines on two different benchmarks and show significant improvements in performance and token efficiency.
- The authors clearly explain the problem with existing methods and motivate the need for a new approach. The ReLoc framework and the revision reward model are explained in a clear and concise way, with helpful figures and algorithms.
- The paper is original in its approach to code generation. While local search as well as reward models to filter candidates have been used across domains, the proposed combination for code generation is novel.
- All experiments are conducted using standard open source datasets and models.

Weaknesses:
- Based on the current set of experiments, it's not clear if the proposed algorithm is equally effective across model families. In particular, do we get similar gains if we use a frontier model like GPT/Sonnet. We expect frontier models to perform well for both initial code generation as well as at revising code based on errors.
- Paper does not provide insights into what kind of errors are recovered during the revision phrase.

---

> ### Author Rebuttal · Authors · 2025-07-31
>
> Thank you very much for your thoughtful and encouraging review. We sincerely appreciate your recognition of the key contributions of our work, including the motivation, design and clarity of the ReLoc framework, as well as the effectiveness of the revision reward model. Below we provide detailed responses to your remaining questions.
>
> > W1 & Q1: Generalization across model families, such as GPT or Claude Sonnet.
>
>
> Thank you for your valuable suggestion. We agree that evaluating the effectiveness of our method across different model families, especially frontier models, is crucial to demonstrate its generality. To show this, we conduct an experiment using `GPT-4o (gpt-4o-2024-1120`) as the *base model* for both code generation and revision. The revision reward model used in this setting was a `Qwen2.5-7B-Instruct`, consistent with the setup described in Section 4.1. We then applied the ReLoc\_HC framework on LiveCodeBench. Here's the result:
>
> | Methods               | Reward function                  | Pass@1 (%) | Tokens (1K) |
> |-----------------------|------------------------|------------|-------------|
> | Plan Search           | Self-evaluation        | 42.7       | 16.7      |
> | BoN                   | Pass Rate              | 41.8       | 12.2      |
> | **ReLoc_HC (Ours)**   | Revision Reward Model  | **44.2**   | **8.7**    |
>
> These results demonstrate that both our ReLoc framework and the revision reward model are highly adaptable across model families, and can be **effectively applied even to strong frontier models** like `GPT-4o`. This highlights their robustness and general applicability in real-world code generation and revision scenarios.
>
> > W1: Paper does not provide insights into what kind of errors are recovered during the revision phrase.
>
>
> Thank you for the insightful question. Based on our analysis of the LiveCodeBench results, we observe that two major categories of errors are effectively corrected by ReLoc during the local search process:
>
> 1. **Partial correctness and incomplete logic.**
>
>    Initial code often passes some public test cases but fails on edge conditions due to narrow or missing logic. Revisions gradually generalize the solution.
>     * Example: A divisor function fails on prime inputs. Revisions add checks and extend logic to handle all cases correctly.
>
> 2. **Runtime and semantic errors surfaced by execution feedback.**
>
>    Explicit errors like index out-of-bound errors, incorrect loop bounds, or type mismatches. These signals are especially helpful because they explicitly point out failure modes, guiding the model toward targeted repairs.
>
>    * Example: A loop with incorrect bounds causes a runtime error. Feedback directs the model to adjust the iteration logic, quickly fixing the issue.
>
>
>
> > Q2: Importance of initial code quality and alternative initialization strategies.
>
> We agree that the quality of the initial code population has a notable impact on the performance of ReLoc. **As shown in Table 3 of the main paper, we have experimented with random sampling as an alternative initialization strategy.** The results show that even with simple sampling, ReLoc remains highly competitive, highlighting its robustness to the choice of initial candidates.
>
>
> | Method          | LiveCodeBench Pass\@1 (%) | LiveCodeBench Tokens (1K) | TACO Pass\@1 (%) | TACO Tokens (1K) |
> | -------------------- | -------------------------------- | -------------------------------- | ----------------------- | ----------------------- |
> | **ReLoc\_HC**        | **38.4**                         | 7.1                              | **13.3**                | 7.6                     |
> | w/ Random Sampling | 36.9                             | 6.5                              | 13.7                    | 7.1                     |
> | **ReLoc\_GA**        | **35.7**                         | 6.8                              | **15.3**                | 7.7                     |
> | w/ Random Sampling | 34.3                             | 4.9                              | 12.9                    | 6.6                     |
>
> **Sensitivity to Seed Code Quantity:**
> We further study how the number of initial seed codes affects performance. As shown below, ReLoc still delivers strong performance even with a single initial code sample, highlighting its ability to iteratively revise and improve suboptimal solutions.
>
> | **Seed Code Number** | **ReLoc\_HC (Pass\@1)** | **ReLoc\_GA (Pass\@1)** |
> | -------------------- | ----------------------- | ----------------------- |
> | 1                    | 35.2                    | 34.1                    |
> | 3                    | 38.4                    | 35.9                    |
> | 5                    | 38.4                    | 35.7                    |
> | 7                    | 38.7                    | 35.4                    |
>
> In summary, while better initial code helps improve performance, **ReLoc does not heavily rely on initialization quality thanks to its iterative improvement nature.**
>
>
> > Q3: Accuracy of reward model on distinguishing correct vs. incorrect code pairs on unseen samples.
>
> We conduct an evaluation on unseen problems, measuring the accuracy of the revision reward model in identifying the correct code from a pair consisting of one correct and one incorrect solution. We compare our model against two strong baselines: the open-source `Skywork-27B model` and the proprietary `GPT-4o`. The results are shown below:
>
> | **Model**                  | **Correct vs. Incorrect Accuracy (%)** |
> | -------------------------- | ------------------------------------- |
> | Skywork-27B   | 53.4                                  |
> | GPT-4o        | 61.8                                  |
> | **Revision Reward Model (Ours)** | **77.2**                              |
>
> **Our revision reward model significantly outperforms both baselines in identifying correct code**, even in challenging cases where the incorrect and correct solutions are highly similar and both reside within the same local neighborhood of a code sample. This highlights the effectiveness of our reward model, which is explicitly trained to leverage the locality structure of code revisions and capture subtle differences that general-purpose LLMs often overlook.
>
>
> > Q4: How did you decide 7k token budget for experiments?
>
> Thank you for the thoughtful question. Controlling the search budget is indeed a critical factor in real-world deployment.
>
> We select a 7K token budget to **reflect realistic resource constraints**, while still enabling effective search. Although ReLoc continues to improve with larger budgets, as shown in Figure 4 where it consistently outperforms baselines as the token budget grows, we intentionally evaluate under a 7K token setting to simulate practical use cases. Scenarios such as code assistants or classroom tools often operate under latency or cost limits, making small-budget inference particularly relevant.
>
> This setup highlights ReLoc’s strength under constrained conditions, while preserving its ability to scale gracefully when more resources are available.
>
> Thank you again for your valuable and encouraging feedback. We will incorporate all these clarifications and improvements into our revision.

---

> > ### Author Response · Authors · 2025-08-06
> >
> > Dear Reviewer,
> >
> > I hope this message finds you well. As the discussion period is nearing its end with **less than two days remaining**, I wanted to ensure we have addressed all your concerns satisfactorily. If there are any additional points or feedback you'd like us to consider, please let us know. Your insights are invaluable to us, and we’re eager to address any remaining issues to improve our work.
> >
> > Thank you for your time and effort in reviewing our paper.

---

### Official Review · Reviewer_sykM · 2025-07-03

**Clarity:** 3
**Significance:** 3
**Originality:** 3
**Rating:** 4
**Confidence:** 4

**Summary:**

This paper proposes a novel decoding strategy for multi-turn code generation by training a reward model based on revision distance. The authors construct a series of search trees and collect the distance from each node to the correct solution as a reward signal that reflects the node's correctness. A reward model trained on this signal is then used to guide tree search during inference.

**Questions:**

After training, what is the prediction accuracy of the reward model on the edit distance metric over the training and validation sets? Providing this information would help assess how the reward signals can be learned effectively.

**Ethical Concerns:**

["NO or VERY MINOR ethics concerns only"]

**Final Justification:**

The authors' additional supplementary experiments (hyperparameters and prediction accuracy) have effectively addressed my concerns. I maintain my positive view of this paper.

**Limitations:**

Yes

**Quality:**

3

**Strengths And Weaknesses:**

**Strengths**

- Using edit distance as a supervisory signal for reward model training is interesting and novel.

- The proposed reward model achieves substantial improvements on benchmarks on LiveCodeBench and TACO, outperforming existing reward models by a significant margin. ReLoc also demonstrates strong scaling behavior with increased computational budget.

- The paper includes a number of illustrative diagrams, which enhance the clarity and readability of the methodology and experimental results.

**Weaknesses**

- In my opinion, the effectiveness of the revision distance-based reward depends on the tree construction algorithm. This raises concerns about its generalization across different tree search strategies. As noted in Line 275, various search algorithms have distinct characteristics and application scenarios. It is unclear whether revision rewards collected using specific neighborhood sampling methods (e.g., hill climbing and genetic algorithms employed in the paper) can be reliably transferred to other search paradigms.

- The paper lacks a hyperparameter sensitivity analysis. Several unique hyperparameters are introduced (e.g., d_max and K), yet it remains unclear how their values affect the final performance.

---

> ### Author Rebuttal · Authors · 2025-07-31
>
> Thank you very much for your detailed and thoughtful review. We truly appreciate you highlighting the strengths of revision-based reward design and the strong empirical results. Regarding your questions, we provide our responses as follows.
>
> > W1: Generalization of the revision reward model across different search strategies.
>
> The core idea of our revision reward model is to effectively differentiate promising incorrect code candidates by comparing their revision distance. As introduced in Line 158 of the main paper, we use **breadth-first search to construct training data** for the revision reward model. This way, we construct comparison pairs within local neighborhoods (parents, children, siblings) of a node in the BFS tree without being tied to any specific search behavior,  which can well capture the relationship between the incumbent code sample and the neighborhood code samples in local search. This design ensures **the learned preferences are transferable across various local search algorithms**.
>
>
> > W2: Lack of hyperparameter sensitivity analysis for key parameters (e.g., d_max and K).
>
> Thank you for raising this important point. We provide a two-part analysis to demonstrate the **robustness** of our method to hyperparameter and data variations.
>
> 1. **The revision reward model is robust to different BFS sampling hyperparameters.**
>
>     We vary the tree depth $d_{\text{max}}$ and branching factor $K$ used to collect training data. As shown below, performance remains stable even with shallower or narrower trees, indicating that our reward model does not require broad exploration to be effective.
>
>     | **Sampling Setting**                  | **ReLoc\_HC (Pass\@1)** |
>     | ------------------------------------- | ----------------------- |
>     | Full scale (`d_max = 5`, `K = 3`)     | **38.4**                |
>     | Shallower tree (`d_max = 3`, `K = 3`) | 38.2                    |
>     | Narrower tree (`d_max = 5`, `K = 1`)  | 37.4                    |
>
>
> 2. **The revision reward model remains effective even with limited training data.**
>
>     We evaluate sample efficiency by downsampling the training set. As shown below, performance drops only slightly even with a quarter of the data, confirming that the reward model benefits from the locality structure and **generalizes well with fewer training pairs**.
>
>     | **Downsampling Strategy on Training set**         | **LiveCodeBench** | **TACO** |
>     | --------------------------------- | ----------------- | -------- |
>     | Random Quarter (218K pairs)       | 38.1              | 13.2     |
>     | LiveCodeBench subset (363K pairs) | 37.9              | 12.4     |
>     | Random Half (436K pairs)          | 38.3              | 13.1     |
>     | Full-scale training (872K pairs)  | **38.4**          | **13.3** |
>
> > Q1: Prediction accuracy of the reward model on edit distance supervision.
>
> Thank you for your valuable suggestion. We agree that evaluating the accuracy of the reward model is essential for understanding its ability to capture fine-grained revision signals. To show this, we evaluate its **pairwise accuracy** on training and validation sets. We compare it with two strong baselines: the open-source model `Skywork-27B` and the proprietary `GPT-4o`. As shown in the table below, **our model significantly outperforms both baselines**:
>
> | **Model**                        | **Pairwise Accuracy (%)** |
> | -------------------------------- | ------------------------- |
> | Skywork-27B (validation)         | 48.6                      |
> | GPT-4o (validation)              | 56.3                      |
> | **Revision Reward (Training)**   | **94.1**                  |
> | **Revision Reward (Validation)** | **72.6**                  |
>
> This result underscores several important observations:
>
> * **Revision-based ranking is a challenging but effective learning task.** Even subtle code edits require fine-grained discrimination beyond generic correctness signals.
> * **Our model generalizes well**, achieving 72.6% accuracy on unseen validation tasks—substantially outperforming general-purpose LLMs like `GPT-4o`.
>
> Together, these results confirm that our reward model effectively captures differences between code candidates, offering reliable guidance for local search where general LLMs fall short.
>
> Thank you again for all your constructive feedback. We will include these clarifications, especially the construction of the reward model data, and updates in our revision.

---

> > ### Author Response · Authors · 2025-08-06
> >
> > Dear Reviewer,
> >
> > I hope this message finds you well. As the discussion period is nearing its end with **less than two days remaining**, I wanted to ensure we have addressed all your concerns satisfactorily. If there are any additional points or feedback you'd like us to consider, please let us know. Your insights are invaluable to us, and we’re eager to address any remaining issues to improve our work.
> >
> > Thank you for your time and effort in reviewing our paper.

---

> > > ### Comment · Reviewer_sykM · 2025-08-08
> > >
> > > Thank you for your detailed response. Your additional supplementary experiments (hyperparameters and prediction accuracy) have effectively addressed my concerns. I maintain my positive view of this paper.

---

> > > > ### Author Response · Authors · 2025-08-09
> > > >
> > > > We appreciate your comments and positive response. Thank you for all the time and efforts in reviewing this paper.

---

### Official Review · Reviewer_U1NN · 2025-07-05

**Clarity:** 3
**Significance:** 2
**Originality:** 2
**Rating:** 4
**Confidence:** 4

**Summary:**

In this work, the authors present ReLoc, a lightweight and unified local search framework for improvement-based code generation with LLMs. Unlike computationally expensive construction-based inference time scaling methods like ToT and MCTS, ReLoc finds high-quality solutions and enjoys the anytime property by exploring a series of local revisions of an established code sample. Besides, compared to the existing improvement-based methods, ReLoc leverages simple yet effective decision rules to navigate the search space. Furthermore, a specialized revision reward model effectively differentiates code samples based on the potential of each code sample being corrected in future steps, which provides fine-grained preferences when the correctness signal is uninformative. Finally, the authors show the flexibility and expressiveness of ReLoc by developing two well-known local search algorithms, i.e., Hill Climbing and Genetic Algorithm.

**Questions:**

see Weaknesses

**Ethical Concerns:**

["NO or VERY MINOR ethics concerns only"]

**Final Justification:**

After reading the rebuttal and other reviews, most of my concerns are addressed.

**Limitations:**

YES

**Quality:**

3

**Strengths And Weaknesses:**

Strengths

1. The authors propose ReLoc, a lightweight and unified local search framework for code generation.

2.  The authors develop a revision reward model trained with pairwise supervisions derived from revision distance comparisons.

3. The authors conduct extensive experimental evaluations on popular code generation benchmarks.

Weaknesses

1. Although the paper includes some ablation results in the table, it would be better to conduct a more thorough ablation study focusing specifically on the local search algorithm and the revised reward model, as these are the core components of the proposed method.

2. Limited baselines.
While the paper compares against several baselines, the field has progressed rapidly, and many relevant works have been published recently (e.g., in 2025 ICLR). It would strengthen the paper to include comparisons with these newer methods and to discuss their differences more thoroughly.

3. Efficiency of the local search algorithm.
The paper claims that the proposed local search algorithm is more efficient, but this is not demonstrated in the experimental results. It would be helpful to include experiments that explicitly evaluate and compare the efficiency of the local search component.

4. Practicality of reward model training.
Training the reward model requires 872K training pairs, which seems impractical for real-world scenarios. It is unclear how the revised reward model performs when only a small number of training examples are available. Additional experiments or discussion on the sample efficiency and robustness of the reward model under limited data would improve the paper.

---

> ### Author Rebuttal · Authors · 2025-07-31
>
> We sincerely appreciate your in-depth and constructive feedback. We are also grateful for kindly pointing out that our method is well-motivated and implies high expandability and flexibility. Regarding your concerns, we provide our responses below:
>
> > W1: Ablation study on the local search algorithm and the revision reward model.
>
> We conduct additional ablations on the local search algorithm and revision reward model to further assess their robustness and effectiveness. Results are summarized below:
>
> 1. **Ablation on Seed Code Number.**
>     To assess the robustness of our local search framework, we vary the number of initial seed code candidates in ReLoc_HC and ReLoc_GA. As shown below, performance slightly drops when using only a single seed due to reduced diversity in the initial population. However, **the overall performance remains strong and stable across different settings**, demonstrating that both variants are highly robust to initialization size.
>
>     | **Seed Code Number** | **ReLoc\_HC (Pass\@1)** | **ReLoc\_GA (Pass\@1)** |
>     | -------------------- | ----------------------- | ----------------------- |
>     | 1                    | 35.2                    | 34.1                    |
>     | 3                    | 38.4                    | 35.9                    |
>     | 5                    | 38.4                    | 35.7                    |
>     | 7                    | 38.7                    | 35.4                    |
>
>
> 2. **Ablation on Revision Reward Model Data Construction.**
>     We evaluate the impact of key parameters in constructing the training data for the revision reward model, including the maximum depth of the code tree (d\_max), the number of revisions per node (K), and the use of locality-based comparison pairs.
>
>     The results show that our reward model is **robust and effective even when using smaller or shallower trees**, indicating strong sample efficiency and low reliance on large-scale supervision. This is largely due to our use of comparison pairs built from local relationships among parent, child, and sibling nodes, which provide meaningful training signals to guide the local search process.
>
>     | **Setting**                         | **ReLoc\_HC (Pass\@1)** |
>     | ----------------------------------- | ----------------------- |
>     | Full setting (d\_max = 5, K = 3)    | 38.4                    |
>     | Shallower tree (d\_max = 3, K = 3)  | 38.2                    |
>     | Fewer revisions (d\_max = 5, K = 1) | 37.4                    |
>
>
> > W2: Limited comparison to recent baselines from ICLR 2025 and other new work.
>
> Thank you for raising this point. We fully agree that including recent baselines is important for a fair and up-to-date evaluation. **We have already incorporated several strong and representative methods from ICLR 2025, ICML 2025, and NAACL 2025 into our experiments**.
>
> These methods are included in **Table 1** of our main results and are clearly categorized by their inference strategy and reward signal.
>
> | **Method**            | **Reward Function** | **LiveCodeBench** | **TACO** |
> | --------------------- | ------------------- | ----------------- | -------- |
> | CodeTree (NAACL’25)   | Self-evaluation     | 27.7              | 8.2      |
> | SFS (ICLR’25)         | Pass rate           | 32.1              | 10.5     |
> | Plan Search (ICLR’25) | Self-evaluation     | 32.7              | 11.2     |
> | ORPS (ICML’25)        | Pass rate           | 28.8              | 9.8      |
> | **ReLoc\_HC (Ours)**  | Revision reward     | **38.4**          | 13.3     |
> | **ReLoc\_GA (Ours)**  | Revision reward     | 35.7              | **15.3** |
>
> \[1] *CodeTree: Agent-guided Tree Search for Code Generation with Large Language Models*, **NAACL 2025**
>
> \[2] *SFS: Smarter Code Space Optimization Improves LLM Inference Scaling*, **ICLR 2025**
>
> \[3] *Planning in Natural Language Improves LLM Search for Code Generation*, **ICLR 2025**
>
> \[4] *Reasoning Through Execution: Unifying Process and Outcome Rewards for Code Generation*, **ICML 2025**
>
> > W3: Lack of empirical evidence demonstrating the efficiency of the proposed local search algorithm.
>
> Thank you for your valuable suggestion. We fully agree that search efficiency is a critical concern, especially under realistic resource constraints. To demonstrate efficiency more concretely, we provide two forms of evidence:
>
> 1. **Efficiency under different token budgets.**
>     As shown in **Figure 4**, ReLoc achieves comparable or better performance than Best-of-N methods with significantly fewer tokens. For example, **ReLoc reaches the same Pass\@1 as BoN under 15K tokens using only 3K tokens**, illustrating its superior token efficiency.
> 2. **Comparison of token usage and performance across methods.**
>     We further report the token consumption and Pass\@1 accuracy of various methods on the LiveCodeBench benchmark. As shown below, **ReLoc strikes the best balance between effectiveness and efficiency**:
>
>     | **Method**           | **Pass\@1 (%)** | **Tokens (1K)** |
>     | -------------------- | --------------- | --------------- |
>     | Plan Search          | 33.8            | 11.6            |
>     | Code Tree            | 29.1            | 15.3            |
>     | TOT                  | 25.4            | 11.9            |
>     | BoN                  | 31.3            | 15.1            |
>     | ORPS                 | 30.0            | 14.7            |
>     | **ReLoc\_HC (Ours)** | **38.4**        | **7.1**         |
>
> Unlike prior methods such as MCTS or agent-based search, which consume large token budgets (Section 1, Line 62), ReLoc uses lightweight decision rules to guide local revisions efficiently. These results confirm that ReLoc achieves state-of-the-art accuracy with significantly fewer tokens, making it well-suited for practical deployment.
>
> > W4: Sample efficiency of the revision reward model under limited training data.
>
> We appreciate the reviewer’s concern regarding the practicality of training the revision reward model with 872K training pairs. We address this issue from both the efficiency of training and the effectiveness under limited data:
>
> 1. **Efficient and Sample-Efficient Training.**
>    While our full-scale training uses 872K training pairs, it requires only **36 GPU hours on 4 H100 GPUs**, which is practical in real-world settings. Importantly, we did not perform any sophisticated data filtering, all sampled data were used. In practice, we found that the revision reward model still exhibits strong performance in combining with ReLoc given fewer samples. We evaluated three downsampling strategies:
>    * **Random Quarter (218K pairs)**
>    * **LiveCodeBench subset (363K pairs, only using data sampled from LiveCodeBench)**
>    * **Random Half (436K pairs)**
>
>    The results on LiveCodeBench and TACO benchmarks (using ReLoc\_HC) show that even with reduced data, the reward model remains effective. This confirms that our revision distance based formulation is **sample-efficient and robust** under data constraints.
>
>     |Downsampling strategy| LiveCodeBench              |TACO|
>     |----------------------|-----------------------|--------|
>     | Random Quarter (218K pairs) | 38.1          | 13.2     |
>     |LiveCodeBench subset(363K pairs)| 37.9  | 12.4 |
>     |Random Half (436K pairs)|38.3 |13.1 |
>     |Full-scale training(872K pairs)| 38.4  |  13.3  |
> 2. **Robustness Across Models.**
>    To further demonstrate the practicality and generalization of the reward model, we trained it using data collected from Qwen-32B and applied it to guide **GPT-4o**, a strong closed-source model. The results are shown below:
>
>     | Methods               | Rew.                  | Pass@1 (%) | Tokens (1K) |
>     |-----------------------|------------------------|------------|-------------|
>     | Plan Search           | self-evaluation        | 42.7       | 16.7      |
>     | BoN                   | Pass Rate              | 41.8       | 12.2      |
>     | **ReLoc_HC (Ours)**   | Revision Reward Model  | **44.2**   | **8.7**    |
>
>
>    Despite being a 7B reward model, it **significantly boosts GPT-4o’s performance** on code tasks, showing strong robustness across model scales and architectures.
>
> Thank you again for your valuable and encouraging feedback. We will incorporate all these clarifications and improvements into our revision.

---

> > ### Author Response · Authors · 2025-08-06
> >
> > Dear Reviewer,
> >
> > I hope this message finds you well. As the discussion period is nearing its end with **less than two days remaining**, I wanted to ensure we have addressed all your concerns satisfactorily. If there are any additional points or feedback you'd like us to consider, please let us know. Your insights are invaluable to us, and we’re eager to address any remaining issues to improve our work.
> >
> > Thank you for your time and effort in reviewing our paper.

---

> > ### Comment · Reviewer_U1NN · 2025-08-08
> >
> > Thanks for the response, which addressed most of my concerns. I will raise my score to 4.

---

> > > ### Author Response · Authors · 2025-08-08
> > >
> > > Thank you for your thoughtful feedback and recommendation for acceptance. We’re glad our rebuttal addressed your concerns.
> > >
> > > We sincerely appreciate your support and encouragement!

---

### Note · Authors · 2025-08-14

Dear Reviewers and ACs,

We sincerely thank all reviewers for their constructive feedback and insightful suggestions. We are encouraged that reviewers recognized the **clear motivation** of ReLoc (U1NN, sykM, n8zP), its strong empirical performance (sykM, n8zP, ZiY2), the **novel perspective from local search** (U1NN, n8zP), and its ability to **facilitate multi-step code generation** (n8zP, ZiY2).

**We have carefully addressed each concern with additional experiments, analyses, and clarifications.** The main clarifications and additional results from the rebuttal are:
1. **Motivation and Conceptual Contribution**
- (sykM, ZiY2) Clarified how ReLoc enables LLMs to efficiently transform incorrect solutions into correct ones via a guided, step-by-step local search process under the supervision of the revision reward model.
- Emphasized novelty in **adapting classical search heuristics to LLM-driven revision** and coupling them with a learnable, domain-specific reward.
2. **Ablation Studies and Component Analysis**
- (U1NN, sykM) Extended ablations on `seed code size`, `search tree depth` (\$d_{max}$), and `branching factor` ($K$), analyzing sensitivity to final performance.
- (ZiY2, n8zP) Clarified revision strategies and initialization methods, showing **robustness across diverse configurations**.
3. **Baselines in of Comparative Evaluation**
- (U1NN, sykM) We have already incorporated comparisons with the latest SOTA methods ReLoc consistently outperforms them. Experiments with GPT-4o further show that ReLoc remains effective even on strong frontier models.
- (ZiY2) Clarified LiveCodeBench improvement from **32.7% to 38.4%**, noting the **revision reward model is a one-time cost**.
4. **Efficiency Considerations and Practical Deployment**
- (U1NN, sykM, ZiY2) Showed strong performance with fewer training pairs, demonstrating **sample efficiency**.
- (U1NN, n8zP, ZiY2) Evaluated ReLoc under realistic token budgets, confirming ReLoc’s advantage.
5. **Reward Model Evaluation and Error Analysis**
- (sykM, n8zP) Added comparisons with `GPT-4o` and `Skywork-27B` demonstrate that the revision reward model achieves higher accuracy in identifying correct and promising code solutions.
- (n8zP) Added qualitative analysis of **errors corrected** during revision.

`During the rebuttal period, we have made our best efforts and shown our utmost sincerity in addressing the reviewers' concerns.` We sincerely appreciate the reviewers' and ACs' time and effort.

---

### Decision · Program_Chairs · 2025-09-17

**Decision:**

Accept (poster)

**Comment:**

- Summary: This paper advances inference time scaling through ReLoc, a local search framework to effectively revise code for code generation tasks. The method encourages exploring local neighborhoods with decision rules. Neighborhood code generation is a key component of the framework and the paper develops a revision reward model to facilitate candidate generation, which is trained with pairwise supervisions based on revision distances and prefers code with smaller revision distances.
- Strengths:
  - The revision reward model provides a novel contribution.
  - The paper is well written.
  - The problem is well motivated, test time scaling for code generation is of interest to the Neurips community and this is a timely contribution.
  - The empirical evaluations are extensive.
- Weaknesses:
  - The significance of the results is debatable. The results lack error bars.
- Suggestions:
  - Authors addressed reviewer’s questions on clarifications regarding revision strategy, computational cost, generalization across models during the rebuttal. I recommend these are reflected in the final manuscript.
- Recommendation:
  - Three reviewers recommended acceptance of the paper, two of whom gave borderline acceptance scores. One reviewer recommended borderline rejection.
  - I recommend acceptance of this paper due to its novel contributions to test time scaling of code generation. I believe this will be an interesting contribution to the field.